# Mothers of Children Diagnosed with ADHD: A Descriptive Study of Maternal Experience during the First Three Years of Treatment

**Pernille Darling Rasmussen** [1,2,*] , **Johanne Pereira Ribeiro** [1] **and Ole Jakob Storebø** [1,3,4]

1   Center for Evidence-Based Psychiatry, Psychiatric Research Unit, Psychiatry Region Zealand, 4200 Slagelse, Denmark; joper@regionsjaelland.dk (J.P.R.); ojst@regionsjaelland.dk (O.J.S.)
2   Private Hospital Hejmdal, 2000 Frederiksberg, Denmark
3   Department of Psychology, University of Southern Denmark, 5000 Odense, Denmark
4   Department of Child and Adolescent Psychiatry, Psychiatry Region Zealand, 4000 Roskilde, Denmark
*   Correspondence: pdra@regionsjaelland.dk

**Abstract:** *Background*: Attention deficit hyperactivity disorder is the most common childhood psychiatric disorder. Current treatment strategies do not provide a convincing improvement on overall functioning, and further, reciprocity between ADHD and attachment has been suggested. This suggests that we do not fully comprehend the mechanisms of the disorder. This study was part of a larger project investigating factors of potential importance when a child is diagnosed with ADHD. *Aim*: In this current study we aimed to gain a clearer understanding about whether the mothers experienced the diagnostic process and treatment as helpful. *Method*: Sixty children newly diagnosed with ADHD and their mothers were included three years prior to this study. Fifty-two (87%) completed a survey about their experience with the diagnostic process and the years after in the psychiatric system and the secondary healthcare sector. Forty-three had also participated in an attachment interview in the original study and answered questions about this. *Discussion*: The follow-up questionnaire was based on conversations with the mothers was not meant to be used as a quantitative measure. However, one point to take is that the mothers did often not feel the help offered to be sufficient. In our opinion, this underlines that we are still far from understanding what ADHD is and what causes the differences in developmental trajectories as well as how differences in etiological factors may call for more customized approaches in treatment strategies.

**Keywords:** ADHD; maternal experience; follow-up; descriptive study

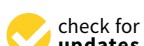



## 1. Background

Attention deficit hyperactivity disorder (ADHD) is the most common childhood psychiatric disorder, and prevalence estimates often exceeds 5% in the general population [1–3]. The condition is characterized by core symptoms such as inattention, lack of concentration, and symptoms of hyperactivity and impulsivity [4]. Problems with emotion regulation are recognized as a consistent trait in children with ADHD [5,6].

Many Cochrane reviews indicate that current treatment strategies do not provide convincing improvement on the overall functioning, which suggests that we do not fully comprehend the mechanisms of the ADHD disorder [7–9]. Importantly, some children diagnosed with ADHD continue to be significantly affected by their disorder as they reach adolescence and adulthood [8]. This increases the risk of serious issues continuing into adulthood despite treatment, which results in increased risk of substance abuse and committing crime as well as a higher mortality rate [10–12]. Additionally, people with ADHD may experience inferior accomplishments on societal parameters, such as occupation and education, as well as experience a higher general morbidity [13–15]. However, one of the few existing long term-studies on children with ADHD found that some children

experience symptom relief to an extent where the disorder seems non-present when they reach adulthood [16]. In the same study, it was also found that developing comorbidity in the form of conduct disorder seemed to predict the adverse trajectory of elevated risky behavior. Why some of the participating children were more likely to develop comorbidities, however, remains unclear. There is an existing possibility that some children displaying ADHD symptoms are primarily affected by inborn neurobiological errors, whereas others might be affected to a higher extent by environmental factors. This hypothesis at least partially explains the differences in treatment outcomes and certainly points to a need for further research. However, overall, we are still far from understanding exactly what predicts the long-term developmental outcomes in ADHD.

It is well known that the general treatment experience can differ significantly from patient to patient [17,18]. The probability exists that the general treatment experience can similarly differ for parents of children receiving psychiatric treatment. In a qualitative study, semi-structured interviews were used to explore potential unmet needs in families with ADHD. Caregivers of children identified a range of issues, such as social impact and strained relationships, that seemed to persist despite treatment, which was, in general, considered to be effective [19]. These findings were supported in a large cross-sectional survey conducted in 10 European countries. Observations were collected from caregivers on their children with ADHD who were currently receiving or had recently received pharmacotherapy. They found significant variations between countries on a range of parameters, such as reported rates of ADHD comorbidity and treatment modalities. Importantly, despite treatment, they found that from the caregiver perspective, the disorder was still associated with a great burden for the family [20]. In the same cohort, it was also found that a third of the caregivers reported great difficulties in obtaining the diagnosis, and more than half reported issues such as lack of resources and gaps in the support provided [21].

*ADHD in a Danish Healthcare System*

Denmark has a decentralized public healthcare system. The Danish public health insurance is mainly financed through general taxation and provides free and equal access to healthcare for all Danish citizens. Denmark consists of five state regions with 98 municipalities in total. The regions are responsible for hospitals, psychiatric services, general practitioners, and specialists (primary health care sector), while the municipalities are responsible for public health, health promotion, and preventive care, children's health services, including dentistry, and elderly care, such as nursing homes, district and home care, and rehabilitation (secondary healthcare sector) [22]. As the responsibility of healthcare is divided in two (the primary healthcare sector driven by the regions and the secondary healthcare sector driven by municipalities), a collaboration between these is crucial for a continuous healthcare experience.

Unfortunately, few studies ask the families affected by ADHD in Denmark about their experiences and what matters to their situation. Laugesen and colleagues (2017) investigated in a qualitative study the parental experiences with the healthcare system after their child had received a diagnosis of ADHD. They found factors, such as trusting relationships and access to health care professionals that recognized their daily struggles, to be of essence [23].

A sound advice often given to medical students comes to mind: when in doubt, ask the patient. With this study, we attempt to elucidate this question. To do so, we followed a group of children and their mothers for three years subsequent to the children's ADHD diagnosis.

## 2. Aim

The aim of this study was to increase our knowledge of the maternal experience of participating in a research project and of help and support in the psychiatric system and the secondary healthcare sector in Denmark for three years subsequent to their child's ADHD diagnosis.

### 3. Method

Sixty children newly diagnosed with ADHD and their mothers were included three years prior to this study in another project that studied factors that could be of importance to treatment outcome [24]. Paternal data was largely excluded in the original collection of data, as less than 25% responded adequately. During the inclusion period, mothers were asked about their experience so far and which questions they would consider important to ask the participating families in a later follow-up study with clinical assessment.

Based on these conversations in the inclusion period, a follow-up questionnaire was created in which mothers were invited to answer questions about their experience of the process. This current manuscript covers results from the short questionnaire focusing on maternal experience.

### 4. Measure

A questionnaire that contained both closed and open-ended questions was used (see Appendix A). Open-ended questions were used to provide an opportunity for the mothers to elaborate on their answers.

To obtain the highest possible response rate, we used the online clinical data managing system Easytrial©, which allowed us to send out gentle reminders. The questionnaire was sent as links to the mothers' e-mails, and responses were retrieved directly to a secure database in compliance with good clinical practice (GCP) and database security legislation.

### 5. Participants

The participants of this study had been included in a previous study on a small cohort of children newly diagnosed with ADHD [25]. There were no further inclusion or exclusion criteria to the follow-up. However, in the original sample, children were included only after attending the Kiddie Schedule for Affective Disorders and Schizophrenia interview (K-SADS) confirming the diagnose of ADHD (314.01) or ADD (314.00) according to DSM-5 (*American Psychiatric Association. (2013). Diagnostic and Statistical Manual of Mental Disorders (DSM-5®)* n.d.). Exclusion criteria were IQ below 70, children suspected of psychosis, or children who were adopted or living in foster care.

At the point of inclusion, the children were aged 7–12 years with a mean age of 9.1 years.

The mothers and children participated in separate attachment interviews, as a central part of the initial study was to investigate the potential influence of current attachment representations and the mother-child relationship. In these interviews, only 15% of the children were found to have a secure attachment, and only 23% of the mothers were classified with a secure attachment representation [26]. Comparably, in the general population, it is expected that 62% of school-aged children and 58% of the mothers will classify as having a secure attachment [27,28].

### 6. Results

All 60 mothers were invited to answer the follow-up questionnaire, and 52 (87%) completed the survey. Gender representation in the original sample of children displayed an expected disproportionate representation of males (72%). This proportion carried over to the follow-up, which was comprised of 37 children and had a male representation of 71%.

The last item in the questionnaire was an open invitation for the mothers to comment on aspects they considered important in their current situation and to participate in a qualitative interview about their subjective experiences of having to care for a child with special needs. Thirty-three mothers (63%) agreed to be contacted for further information about the qualitative interview. They all gave consent to be interviewed after they had received this information. The main reason for participation in the interview given by the mothers was a wish to contribute to research with their personal experience and knowledge in order to help other families in similar situations.

Demographic data (Table 1) displayed that the majority of parents had full-time jobs. However, when combining the categories "unemployed" and "on sick leave," many were unemployed compared to the general population. A total of 23% of the mothers and 18% of the fathers were not working, whereas this was only the case for 3.7% of the general population in January 2019 [29]. Educational levels of the parents are presented in Table 1. The majority of mothers (73%) had an educational level corresponding to a two-year, professionally oriented higher education or lower, and only 4% had a higher education corresponding to an academic master's degree. A total of 82% of the fathers had an educational level corresponding to a vocational degree or lower.

**Table 1.** Parental demographics and civil status.

| | Mothers n = 52 (%) | Fathers n = 50 (%) |
|---|---|---|
| Employment status | | |
| Full-time job | 33 (63.5%) | 40 (80%) |
| Part-time job | 7 (13.5%) | 1 (2%) |
| Unemployed | 9 (17.3%) | 5 (10%) |
| Sick leave | 3 (5.7%) | 4 (8%) |
| Not relevant | 0 | 2 |
| Level of Education | | |
| Unskilled worker | 9 (17.3%) | 11 (22%) |
| Skilled worker | 13 (25%) | 20 (40%) |
| Secondary education or equivalent | 16 (30.8%) | 10 (20%) |
| Medium tertiary education (e.g., bachelor's) | 12 (23.1%) | 8 (16%) |
| Long tertiary education (e.g., graduate from university) | 2 (3.8%) | 1 (2%) |
| Not relevant | 0 | 2 |
| Changes in civil status since inclusion three years prior | | |
| Divorced—unchanged | 26 | - |
| Married—unchanged | 22 | - |
| Divorced—changed | 2 | - |
| Married—changed | 0 | - |
| Not relevant | 2 | - |

Note that, in the tables, when denoting "not relevant," this corresponds to the two children who had no contact with a father. The mothers had little or no information about the fathers, which is why to the sample of fathers is limited to n = 50.

Two couples were divorced after the initial assessment, which left the total percentage of single-parent families to be 58%. In all cases of single-parent families, the child's address remained with the mother. Compared with the general Danish population in 2019, 73% of children lived with both of their parents and 27% in single-parent families.

In Table 2, the mothers gave information on any changes in the diagnosis since the first diagnostic process. Thirteen of the children (25%) had received a reassessment due to lack of improvement or worsening of their symptoms. Out of the 13 children, two had the original diagnosis confirmed and received no additional diagnoses. The remaining 11 children received an additional diagnosis: three children received a diagnosis of autism; four were diagnosed with conduct disorder; one with Tourette's syndrome and OCD; and one with a specific learning disability. Two of the children were sent to pediatric assessment, where one was diagnosed with epilepsy, and the other was diagnosed with probable mild brain damage due to head trauma at the age of three. This development points to the circumstances of progress in the years succeeding the first ADHD diagnosis. It indicates that ADHD may be a rather unspecific diagnostic category and confirms the diversity in how disorder characteristics manifest and emerge in the individual.

**Table 2.** Information related to the diagnostic process.

| | Mothers (n = 52) |
|---|---|
| Reassessment of diagnosis since entering the project | |
| Yes | 13 (25%) |
| No | 39 (75%) |
| Remaining contact with the regional child and adolescent psychiatry | |
| Yes | 35 (67.3%) |
| No | 17 (32.7%) |
| Change in level of functioning | |
| No change | 11 (21.2%) |
| Worsening since original diagnostic assessment | 6 (11.5%) |
| Increased functioning since original diagnostic assessment | 35 (67.3%) |

Thirty-five (67%) of the children still had regular contact with departments of child and adolescent psychiatry. This was mainly due to regular assessment of the effects of medical treatment.

The mothers were asked to rate any changes in their child's level of functioning since they had received a diagnosis. A total of 67% had an increase in general functioning, whereas 21% had no change. Six of the mothers (11%) indicated worsening in the level of functioning.

In Table 3, the mothers were asked to rate the significance of a number of factors in the time after the initial diagnostic process. Regarding medical treatment, eight children (15%) did not receive any kind of medication at the point of follow up. Thirty-six of the mothers (70%) perceived the medical treatment to have either some or great influence, whereas eight of the mothers did not see any difference. When excluding the children that did not receive medical treatment, the number of mothers indicating positive effects of the medicine increased to 82%. However, as six of these children had at some point received medication, the total number of mothers that did not see any significant effect is more likely to be 14 (27%). We did, however, not ask whether they discontinued the treatment due to small effects or side effects.

**Table 3.** Maternal assessment of factors of importance in the treatment.

| | Mothers (n = 52) |
|---|---|
| Significance of medical treatment | |
| Little or no influence | 8 (15%) |
| Some or great influence | 36 (70%) |
| Not relevant | 8 (15%) |
| Significance of increased support in school or/and at home | |
| Little or no influence | 22 (42%) |
| Some or great influence | 18 (35%) |
| Not relevant | 12 (23%) |
| Significance of maternal psychoeducation (learning about the diagnosis) | |
| Little or no influence | 25 (48%) |
| Some or great influence | 27 (52%) |
| Not relevant | 0 (0%) |

Not relevant = non received.

According to the National Clinical Guidelines, it is considered appropriate to initiate treatment with non-pharmacological interventions as first-line treatment [30,31]. It is recommended that medical treatment be used only when educational and psychological approaches have been shown to be inadequate in obtaining symptom control [30].

Twenty-two of the mothers (42%) indicated that increased support in school or/and at home had little or no influence on general functioning, including ADHD symptoms. When leaving out the 12 mothers who indicated that no measures were taken in the secondary

health care sector yet, more than half of the mothers (55%) indicated that they experienced no effects of the support from the secondary healthcare sector. There could be a discrepancy between maternal expectations and reality, albeit the numbers suggest a need to evaluate the support offered to caregivers of children with ADHD. There is an obligation for the municipalities to follow the Danish Health Care Act, which sets the bar for health care but is up the individual municipality as to how they want to allocate their resources as well as what comprises the general support. This means that general support, to some degree, can vary in quality and extent from municipality to municipality.

Finally, the mothers were asked to assess their benefits of psychoeducation. This is the only parameter that all the mothers agreed to have received. Psychoeducation is usually offered to all families by the departments of child and adolescent psychiatry immediately after a diagnosis is given. A little more than half (52%) considered this to be of some or great relevance.

In the part of the questionnaire that focused on aspects of significance in treatment, the mothers could comment openly about the aspects they considered significant to any improvement or lack of improvement in the time subsequent to the ADHD diagnosis of their child. This was done as an acknowledgement of the obvious limitations of asking prepared and close-ended questions. What we as researchers and clinicians assume to be of great importance might not resonate with and correspond to the perceptions of the families.

Thirty-two of the mothers had no additional comments. Of the remaining 20 mothers, various aspects were mentioned, but generally, two themes emerged. Primarily, they acknowledged the relief for the child itself when he or she learned about the diagnosis and realized that there was an explanation for the difficulties they had experienced. Secondly, the mothers experienced an improvement in general functioning after the child was offered a placement in a special needs class with fewer students in each class, more teachers with knowledge on ADHD available per class, as well as increased learning resources.

In the final part of the questionnaire, we asked about the maternal participation in the Adult Attachment Interview (AAI) (Table 4). This should not, per se, have an influence on the child's functioning; however, as attachment problems have been found in the majority of children with ADHD, and further intergenerational relay is found in non-clinical populations, we decided to include questions on this part, too. The clinicians conducting the interviews expressed great surprise about the findings, as an unexpected number of maternal narratives carried severe traumatic experiences. This phenomenon is elaborated in an upcoming paper focusing on the attachment interviews.

Forty-three mothers participated in both the attachment interviews and responded to the follow-up questions. Twenty-seven of these (62%) stated that they clearly remembered the interview three years in retrospect. When asked about potential effects of participating in the interview, 26 (96%) stated that the interview brought on reflections of their childhood, and 21 (78%) that it had initiated further reflections on their own role in parenting. As the interview digs into early childhood memories to activate the attachment system, it is not very surprising that the mothers who had clear memories were also the ones who had the most reflections about their childhoods. One can wonder if the ones stating that they did not remember much did not remember as an expression of a defense mechanism taking place. It is well known that the interview is designed to activate the attachment system [32]. At the moment, however, the question of whether or not remembering is in fact an expression of a defense mechanism is confined to a speculative level.

In a small cohort of children with ADHD, it was found that the mothers displayed remarkably high frequencies of insecure attachment. Moreover, the most predominant attachment representation was the dismissing type, characterized by strategies of denial and idealization. Adults characterized as dismissing of attachment often appear as unable or unwilling to take attachment issues seriously. In attachment interviews, they frequently answer questions in a guarded way and avoid elaboration. Furthermore, a consistent feature is that they seem to dislike and distrust looking inwards. People with dismissing

attachment representation often speak vaguely about their parents, with a tendency to describe them in idealized terms. However, when pressed for incidents to support such descriptions, they often display remarkable problems in remembering their childhoods or the memories contradict their general assessments.

**Table 4.** Information on maternal participation in attachment interviews.

|  | **Mothers (n = 52)** |
| --- | --- |
| Participation in attachment interview |  |
| Did not participate | 9 (17%) |
| Participated but remember only a little about it | 16 (31%) |
| Participated and remember clearly | 27 (52%) |
| The interview brought on reflections about my own childhood |  |
| Not at all or only a little | 8 (16%) |
| Do not know | 9 (17%) |
| Yes—some or a lot | 26 (50%) |
| Not relevant (did not participate) | 9 (17%) |
| The interview brought on reflections about my parenting role |  |
| Not at all or only a little | 11 (21%) |
| Do not know | 11 (21%) |
| Yes | 21 (40%) |
| Not relevant (did not participate) | 9 (18%) |
| The interview brought on reflections about my parenting role |  |
| Not at all or only a little | 11 (21%) |
| Do not know | 11 (21%) |
| Yes | 21 (40%) |
| Not relevant (did not participate) | 9 (18%) |

## 7. The Final Open-Ended Question

Apart from the possibility to elaborate on own experiences, the mothers were given the opportunity to express anything they considered to be of importance as the final question.

It was not the intention for this item to be a part of the dataset but merely an opportunity for the mothers to describe how they experienced the encounter with the psychiatric system both in the primary and secondary healthcare sector, as we suspected these answers to be useful in generating and qualifying new hypotheses.

After reading the answers, however, we decided to report a few of the statements here, as they express some of the frustrations that parents of children with ADHD experience and can point to areas that need more attention. As an example, one mother wrote:

> "Please, would someone just write up a manual on how to get help from the system? What can I expect, and what am I supposed to just handle on my own?"

Another mother wrote:

> "It feels like it is impossible to get any help if you do not want to medicate your child. I do not want medication! I want someone to help me to help my daughter. Why is that not part of the treatment?"

Both mothers expressed what several others exclaimed: that they feel abandoned once the diagnostic process is over and that they have to fight the system to get help and even more so if they dismiss pharmacological treatment for their child.

A mother reflected on her participation in the attachment interview:

> "It dawned on me during that interview. I am still affected by my childhood, and I am bringing these experiences into my parenting. I went back to my parents to talk about it, and it made a huge difference in how I see myself as a parent and a human being in general. I am doing this as best as I can."

The statements above raise important questions:

Do we know enough about what is important for each family? Do we assume too fast or falsely that the things health professionals consider helpful are, in fact, also what the families find helpful?

Is there a need for more focus on the transition between the two healthcare sectors? Do we keep in mind that parents possess knowledge of their own child but do not have the same preconception as clinicians on how the system works?

## 8. Perspectives and Limitations (Concluding Remarks)

This study can be seen as an attempt to qualify future hypotheses on ADHD by adding the aspect of maternal experience of the current diagnostic process and treatment strategies. It is a small contribution and is to be considered as merely an attempt to initiate further reflection on how we develop our knowledge in the field of treating ADHD. In our opinion, we are still far from understanding what ADHD is and what causes the differences in developmental trajectories as well as how differences in etiological factors may call for more customized approaches in treatment strategies. Etiologically ADHD is a complex condition with known genetic and hereditary aspects [33–35]. However, environmental factors have similarly proven to be of importance in the development of the disorder [34,36,37], e.g., a mutual relationship has been found to exist between insecure attachment and ADHD [38]. The nature and clinical significance of this reciprocity has been difficult to establish although it has been suggested that the common denominator is difficulties in emotion regulation [39]. Another study supports the notion that the general level of resources in the family may be a factor of importance that should be taken into consideration [40]. As such, it has been found that parents of children with ADHD had an increased rate of psychopathology and that more than 50% of parents display high scores on ADHD-symptoms without being diagnosed [41–43]. These findings point to the fact that hereditary and environmental factors are not mutually exclusive. It further indicates that more research is needed if we are to understand the aspect of the burden that families experience when having a child with ADHD.

The interviewers in the initial study returned with intriguing new insights on the mothers. Furthermore, some of the mothers themselves expressed their wish for psychiatric research to extend the focus to the whole family or at least areas other than just the child's current symptom score. In this small study, we cannot distinguish the ones that experienced great improvement from the ones who did not. However, it is worth noting that 27% of the mothers stated that there was no improvement in their child's daily functioning after three years of treatment. As they all received the same offer, their vastly different experiences and perceptions of the usefulness of treatment points to the need for a more individualized treatment offer. This, again, may indicate the need to know more about the families than just the child's current symptom score in order to create an effective stepped-care model [44].

There are obvious limitations to this study. Firstly, we did not use a validated questionnaire. The questionnaire was based on conversations with the mothers in the inclusion period; hence, there is an existing possibility that the mothers could have developed a different view on what would be important to ask three years later. Further, the sample size is very small. However, the response rate was high, which indicates that the mothers found it meaningful to participate. As we set out to gain knowledge on the maternal experience, this is a good indicator of the aim of the study. The mothers were screened in the original study, and none had a clinical diagnose of ADHD. However, when screening for ADHD-symptoms, perceived attachment style, and resilience, we found a potential link indicating that higher scores on the Adult ADHD Self-report Scale (ASRS-v1.1) were associated with lower scores on self-reported resilience [40]. This indicates that even factors such as subclinical ADHD-symptoms may be of importance. We did not include this perspective in the current study.

Lastly, it should be noted that the fathers did not receive this questionnaire.

### 9. Conclusions

Few conclusions can be drawn from this study; however, we hope to raise awareness on an area that is largely unexplored in ADHD research. The consideration to gain from this study is that we may not have a clear understanding of what families consider to be the greatest stressors in ADHD treatment and what is more helpful in the time after the diagnosis. This indicates that we may have a lack of important information when planning future research and developing new treatment strategies. There is a general need for a more thorough mapping of characteristics, including resources and vulnerabilities in the families around ADHD children. It will be a large and extensive task, but in due course, a more thorough description of the individual family may lead to a more personalized treatment strategy, which in turn can improve the prognosis for each individual child with ADHD.

**Author Contributions:** Conceptualization P.D.R. and O.J.S.; project administration and data curation, P.D.R.; writing—original draft preparation P.D.R.; writing—review and editing, J.P.R. and O.J.S.; All authors have read and agreed to the published version of the manuscript.

**Funding:** This research received no external funding.

**Institutional Review Board Statement:** Not applicable.

**Informed Consent Statement:** Informed consent was obtained from all subjects involved in the study.

**Conflicts of Interest:** The authors declare no conflict of interest.

### Appendix A. Questionnaire

**Introduction: Some children have their diagnosis reassessed after some time in treatment. This is the focus of the first part of the questionnaire:**

1. **Which diagnosis(s) did your child receive after the initial psychiatric assessment? (tick off in "unknown" if you do not remember the exact diagnosis)**

   0 = none
   1 = ADHD
   2 = ADD
   3 = Infantile Autism
   4 = Asperger's Syndrome
   5 = Atypical autism
   6 = OCD
   7 = Anxiety
   8 = other

2. **Has there been a reassessment of diagnosis since?**

   0 = No
   1 = Yes

3. **If so, please indicate which diagnoses has been added or removed (reply "0" if there are no changes and tick off in "unknown" if you do not remember the diagnosis)**

   **(Open field for reply)**

4. **I will now ask you to make a general assessment on how well you feel your son/daughter is doing now compared to when you were first referred to psychiatric evaluation?**

   1 = Worsening
   2 = No change
   3 = Increased functioning

5.   **Do you have a remained contact with child and adolescent psychiatry?**

   0 = No
   1 = Yes

   **Questions about the post-diagnostic progress:**

6.   **Which of the treatment strategies mentioned below have you been offered and how did they impact your son/daughter's current well-being?**

   **Significance of medical treatment**

   1 = Little or no influence
   2 = Some or great influence
   3 = Not applicable (not offered/accepted)

   **Increased support in school and in home (in municipal settings)**

   1 = Little or no influence
   2 = Some or great influence
   3 = Not applicable (not offered/accepted)

   **Psychoeducation (learning about the diagnosis and how to handle it)**

   1 = Little or no influence
   2 = Some or great influence
   3 = Not applicable (not offered/accepted)

   **Any other actions affecting current well-being:**

   Please indicate if there have been any other factors that you think have made a difference in either positive or negative direction (type "0" if there is nothing to add):
   **(Open field for reply)**

   **Questions about maternal participation in the Adult Attachment Interview (AAI):**

   **Some of you participated in an interview about your own upbringing and attachment experiences, for which we have a few questions.**

7.   **Did you participate and do you have a clear memory of the interview?**

   0 = I did not participate in the interview
   1 = I participated but do not remember the interview
   2 = I participated and remember the interview

8.   **Answers below to how well you think the inserted statements fit on you:**

   **The interview brought on reflections about my childhood and how it has affected my adulthood**

   0 = Not at all or only a little
   1 = Do not know
   2 = Yes – some or a lot
   3 = Not relevant (did not participate)

   **The interview brought on reflections about how my upbringing has affected my parenting**

   0 = Not at all or only a little
   1 = Do not know
   2 = Yes – some or a lot
   3 = Not relevant (did not participate)

   **The last questions screen for any changes in domestic conditions:**

9.   **What is your current occupation?**

0 = Full-time employee
1 = Part-time employee
2 = unemployed
3 = Sick leave
4 = Not relevant

10.  **What is your current basic level of education?**

0 = Unskilled worker
1 = Skilled worker
2 = Secondary education or equivalent
3 = Medium tertiary education (e.g., bachelor)
4 = Long tertiary education (e.g., graduate from university)

11.  **What is the current occupation of the father to your child?**

0 = Full-time employee
1 = Part-time employee
2 = Unemployed
3 = Sick leave
4 = Not relevant

12.  **What is the current basic educational level of your child's father?**

0 = Unskilled worker
1 = Skilled worker
2 = Secondary education or equivalent
3 = Medium tertiary education (e.g., bachelor)
4 = Long tertiary education (e.g., graduate from university)

13.  **Are there changes to your civil status?**

0 = Unchanged since last contact - married/cohabiting with father to son/daughter
1 = Unchanged since last contact - divorced from father to son/daughter
2 = Not applicable (e.g., if there has never been contact between father and child)
3 = Changed since last contact - divorced from father to son/daughter
4 = Changed since last contact - married to father to son/daughter

14.  **Is there anything else you think is essential to let the professionals know?**

(It can be about anything you consider to be important, and type "0" if you have nothing to add)
**(Open field for reply)**

15.  **As part of the project, we would like to invite you to an in-depth interview about how you have experienced the process of having your child diagnosed (and the time before and after). Please tick if you would like to participate in this.**

0 = no
1 = yes

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
