# Peer review of "Mothers of Children Diagnosed with ADHD: A Descriptive Study of Maternal Experience during the First Three Years of Treatment"

_pediatrrep, doi:10.3390/pediatric13030052_

Round 1

Reviewer 1 Report

The study is a qualitative assessment of the experience of mothers whose children have been diagnosed and treated fo ADHD. While this in general is a topic of high interest, the methodology allows not to draw strong conclusions and the study design has severe limitations. 

  1. Abstract: The authors should give results in the "results" section and not give the numbers of mothers who particiated which belongs more in the method section
  2. What exactly is the aim of the study? To what should increased knowledge of the maternal experience then lead? 
  3. Did the authors at least screen the mothers for an adult ADHD diagnosis or history of childhood diagnosis in the mothers?
  4. Maternal ADHD could explain the high load of maternal trauma experience. ADHD patients have an increased risk of experiencing trauma. 
  5. To draw valid conclusions from the Attachment questionnaires, also the psychopathology of mothers should be asssessed in detail as maternal mental disorders strongly influence attachment to the child. 
  6. Minor points: The tables could be merged into few tables. 

Reviewer 2 Report

General comment: 

The manuscript entitled „Mothers of Children diagnosed with ADHD: A Descriptive Study of maternal experience during the first three years of treatment.” describes maternal experiences of treatment sufficiency after their child has been diagnosed with Attention Deficit/Hyperactivity Disorder. Thereby the manuscript addresses an important topic and explores potential unmet needs within the Danish health care system.

While the manuscript is informative about maternal experiences, I would, however, recommend rephrasing some aspects within the abstract. Since the maternal experiences were not related to any child behavior, it is currently only speculative to say that this might be due to differences in etiological factors (at least other than the attachment styles and comorbidities that were identified later as described within the manuscript). Albeit likely being the case and as mentioned in the conclusion section of the main manuscript, we cannot draw any further conclusions on this issue based on the presented data. With regard to the nature of the study, one may thus change the phrasing of some aspects within the abstract so that it does not suggest that the causes of ADHD and the origin of different trajectories is addressed within the study, but instead, as the last sentence within the abstract stresses, gain a clearer understanding about whether the families experience the treatment as helpful.

Background:  

  1. Could the authors please describe to the readers what is new in this study compared to the study by Laugesen et al. 2017 that is mentioned within the section on the Danish health care system?

Methods:

  1. Unfortunately, I was not able to see any supplementary material to see further information on the questionnaire: Did any question inquire whether mothers have anything / any area they would particularly like more support with?
  2. Participants: Do the authors have any other demographic information on the mothers, e.g., current age of the mother/age at birth?

Results:

  1. I would advise revising the section in which the experience of medical treatment is addressed. This section is currently somewhat difficult to read. One might consider describing the findings of maternal experiences of children that received medical treatment only. The ratings of maternal experiences with their children that did not receive any medical treatment are hard to interpret and it is questionable as to what has been rated.

  1. After the National Clinical Guidelines are mentioned it is described that the support in school and/or home had little effect. Could the authors please elaborate what kind of support can be provided to the children at school or home with the Danish healthcare system? Does this refer to special need classes with fewer students and more teachers that are mentioned later in the paragraph on the aspects of significance in treatment?

  1. When describing the upcoming paper on severe traumatic experiences: The authors might consider uploading their manuscript on a preprint server to accurately refer to it within the current manuscript.

Limitations and perspectives (concluding remarks)

  1. Minor note: better use “perspective and limitations” instead of “limitations and perspectives” since the limitations follow the perspectives within the main text as well.

Discussion:

  1. It would be interesting to examine whether, in particular (or only?), those mothers whose children received additional diagnoses were the ones who reported that there was no improvement in their child's daily functioning. It could be that mainly due to the high comorbidities in this sample, the initial treatment of a child with only ADHD was rated as not sufficient because the treatment was not yet adapted to the needs arising from the comorbid diagnoses.

Round 2

Reviewer 1 Report

The authors improved the abstract and answered my questions. However, I suggest they also add in the manuscript that 

 - non of the mothers had a diagnosed ADHD and they should add the instrument with which the mothers were screened (ASRS?) 

 - and the mothers should also have been screened for other mental disorders using for example SKID-I/II, if this was not the case it should be mentioned as a limitation also in the manuscript 

Table 1a-1c could be easily just one bigger table instead of three tiny tables. 

 - 

Author Response

The authors improved the abstract and answered my questions. However, I suggest they also add in the manuscript that 

 - non of the mothers had a diagnosed ADHD and they should add the instrument with which the mothers were screened (ASRS?) 

 - and the mothers should also have been screened for other mental disorders using for example SKID-I/II, if this was not the case it should be mentioned as a limitation also in the manuscript 

Table 1a-1c could be easily just one bigger table instead of three tiny tables. 

Response:

Thank you to the reviewer for your time and valuable inputs! We have added the requested information as a limitation to the study and collapsed the tables as suggested.